# An Evaluation of the Anticancer Properties of SYA014, a Homopiperazine-Oxime Analog of Haloperidol in Triple Negative Breast Cancer Cells

**DOI:** 10.3390/cancers14246047

**Published:** 2022-12-08

**Authors:** Gladys M. Asong, Chandrashekhar Voshavar, Felix Amissah, Barbara Bricker, Nazarius S. Lamango, Seth Y. Ablordeppey

**Affiliations:** 1College of Pharmacy and Pharmaceutical Sciences, Institute of Public Health, Florida A&M University, Tallahassee, FL 32307, USA; 2College of Pharmacy, Ferris State University, Big Rapids, MI 49307, USA

**Keywords:** anticancer activity, sigma-2 receptor selective ligand, triple negative breast cancer, MDA-MB-231 cells, MDA-MB-468 cells, cell proliferation, caspase activation, apoptosis

## Abstract

**Simple Summary:**

Triple negative breast cancer (TNBC) is one of the most challenging and hard to treat breast cancer types due to the lack of the receptor targets that are commonly found in breast cancers. It is aggressive in nature with a high recurrence rate. Chemotherapy, radiotherapy, and surgery are the available options. However, conventional chemotherapy drugs have not been effective and new treatment strategies are needed. To this end, we tested SYA014, a compound that binds to the sigma-2 receptor, for its anticancer properties against two different TNBC cell lines. SYA014 was able to kill the cancer cells by interfering with key cellular events associated with cell proliferation. SYA014 induced cytotoxicity in the TNBC cell lines and was evaluated for the involvement of the sigma-2 receptor. Finally, SYA014 was tested in combination with the well-known anticancer drug, cisplatin, in both TNBC cells and non-cancer cells, to improve the treatment outcome.

**Abstract:**

Triple negative breast cancer (TNBC) is a type of breast cancer associated with early metastasis, poor prognosis, high relapse rates, and mortality. Previously, we demonstrated that SYA013, a selective σ2RL, could inhibit cell proliferation, suppress migration, reduce invasion, and induce mitochondria-mediated apoptosis in MDA-MB-231 cell lines, although we were unable to demonstrate the direct involvement of sigma receptors. This study aimed to determine the anticancer properties and mechanisms of action of SYA014, [4-(4-(4-chlorophenyl)-1,4-diazepan-1-yl)-1-(4-fluorophenyl)butan-1-one oxime], an oxime analogue of SYA013, the contribution of its sigma-2 receptor (σ2R) binding, and its possible synergistic use with cisplatin to improve anticancer properties in two TNBC cell lines, MDA-MB-231 (Caucasian) and MDA-MB-468 (Black). In the present investigation, we have shown that SYA014 displays anticancer properties against cell proliferation, survival, metastasis and apoptosis in the two TNBC cell lines. Furthermore, a mechanistic investigation was conducted to identify the apoptotic pathway by which SYA014 induces cell death in MDA-MB-231 cells. Since SYA014 has a higher binding affinity for σ2R compared to σ1R, we tested the role of σ2R on the antiproliferative property of SYA014 with a σ2R blockade. We also attempted to evaluate the combination effect of SYA014 with cisplatin in TNBC cells.

## 1. Introduction

Breast cancer is one of the most prevalent and common malignancies among women worldwide and continues to be the second-leading cause of death in women. According to Global Cancer Statistics 2020, 2.3 million women were diagnosed with breast cancer and 685,000 deaths were recorded globally in 2020 [1]. It is estimated that 287,850 new breast cancer cases with 43,250 cancer deaths are expected to occur in 2022 in the United States alone [2]. The etiology of breast cancer is complex and still unclear. However, advancement in breast cancer research for the past decade indicated the involvement of five important factors: aging, family history, reproductive factors, estrogen and lifestyle [3]. Among the various subtypes of breast cancers, triple negative breast cancer (TNBC) is an aggressive subtype with poor prognosis compared to other breast cancer subtypes, accounting for 10−15% of all breast cancer cases. TNBC is heterogeneous in nature with a lack of estrogen receptor (ER), progesterone receptor (PR), and human epidermal growth factor receptor type 2 (HER2) expression [4]. Current treatment options for breast cancer include surgery, chemotherapy, radiotherapy, endocrine therapy, and targeted therapy. However, there are limited treatment options for TNBC, as it is insensitive to chemotherapy, endocrine therapy, and targeted therapies, due mainly to a lack of specific receptors [5]. Drug resistance development over time with standard chemotherapeutic agents and a high relapse rate with such treatments also mitigate progress. To address these issues, the development of novel agents for use in monotherapies, along with combination therapies involving standard chemotherapy drugs and new agents, are also being widely investigated [6,7].

Haloperidol, an antipsychotic drug used for the treatment of schizophrenia, psychosis, and bipolar disorders, was reported to show anticancer properties [8,9,10,11,12]. Haloperidol is also known to exhibit non-selective interaction with sigma (σ) receptors (σ1 and σ2) [13] and, as such, it has been speculated that it acts through sigma receptors (σRs). In our search for novel sigma-2 receptor ligands (σ2RLs), we have identified two σ2R selective ligands, SYA013 and SYA014, which are structural modifications of haloperidol. SYA013 is a homopiperazine analog in which piperidin-4-ol of haloperidol is replaced, while SYA014 is an oxime derivative of SYA013 (Figure 1). Similar to haloperidol, SYA013 also binds to σRs. However, unlike haloperidol, SYA013 and SYA014 exhibit a 4.3 and 1.5 fold preferential binding affinity towards σ2Rs over σ1Rs, respectively [14]. Our earlier studies revealed the cytotoxic effects of the compounds SYA013 and SYA014 in solid tumor cell lines such as MDA-MB-468, MDA-MB-231 (TNBC), A549 cells (human alveolar basal epithelial adenocarcinoma), PC-3 (prostate cancer), and MIA PaCa-2 and Panc-1 (pancreatic cancer). These compounds were also evaluated for their effects on the viability of MCF-10A cells (non-tumorigenic breast epithelial cells) in order to evaluate their selective cytotoxicities to proliferative cells [14]. In addition, we have recently reported on the anticancer effects and mechanisms of the action of SYA013 against TNBC cell lines (MDA-MB-231 and MDA-MB-468) [15].

As a continuation of our previous findings, we now explore the anticancer properties of SYA014 against the two TNBC cell lines, MDA-MB-231 and MDA-MB-468, and report its effects on the proliferation, migration, and invasion of TNBC cells. We also attempted to decipher the mechanism/pathways involved in the antiproliferative properties of SYA014 and the role of the σ2R on the antiproliferative effects of SYA014 by blocking the σ2 receptor with RHM-1 (a non-cytotoxic selective σ2R antagonist) [16]. In addition, we evaluated the combination effect of SYA014 with cisplatin for a possible synergistic anticancer effect on these TNBC (MDA-MB-231 and MDA-MB-468) and MCF10A cell lines. Herein, we present the results of a mechanistic investigation into the anticancer properties of SYA014 on TNBC cell lines, MDA-MB-231 and MDA-MB-468 for possible use in monotherapy or in combination with a standard chemotherapeutic drug, cisplatin.

## 2. Materials and Methods

### 2.1. Materials

Human adenocarcinoma breast cancer cells (MDA-MB-231 cells, MDA-MB-468 cells, and MCF-10A) were purchased from American Type Culture Collection (ATCC, Manassas, VA, USA). Cell culture media (Dulbecco’s modified eagle medium, high glucose, GlutaMax DMEM), fetal bovine serum (FBS), and antibiotic mixture solution (penicillin−streptomycin-neomycin) were purchased from Life Technologies (Grand Island, NY, USA). Phosphate buffered saline was purchased from Genesee Scientific (San Diego, CA, USA). µ-Slide 8-well (glass bottom) and cell culture inserts were purchased from ibidi (Madison, WI, USA). Caspase assay kits (Caspase Glo 3/7, Caspase Glo 8, Caspase Glo 9) were purchased from Promega Corporation (Madison, WI, USA). ApopNexin Annexin V FITC Apoptosis Kit, cell cycle analysis kit and bioreagents (ethidium bromide and acridine orange) were purchased from Sigma-Aldrich (St. Louis, MO, USA). Apoptosis antibody sampler kit, proapoptosis Bcl-2 family antibody sampler kit, and cytochrome c antibody kit were purchased from Cell Signaling Technology (Danvers, MA, USA). Mini-PROTEAN TGX precast protein gel, Precision Plus protein dual color standards, Immun-Blot PVDF membrane, Laemmli premixed protein sample buffer for SDS-PAGE, tris buffered saline, 10× tris/glycine premixed electrophoresis buffer, blotting-grade blocker nonfat dry milk for western blotting, and clarity western ECL substrate were purchased from Bio-Rad (Hercules, CA, USA). Siramesine and PB28 were purchased from Millipore Sigma (St. Louis, MO, USA). SYA014 was synthesized and characterized in our lab as reported previously [14]. RHM-1 was synthesized as per the reported procedure elsewhere [16].

### 2.2. Cell Culture

MDA-MB-231 and MDA-MB-468 were cultured in Dulbecco’s modified eagle medium high glucose, GlutaMax (DMEM). The media were supplemented with 10% fetal bovine serum (FBS) and PSN (100 U/mL penicillin, 50 µg/mL neomycin). MCF-10 cells were maintained in DMEM/F-12, supplemented with insulin (10 µg/mL), epidermal growth factor (EGF, 20 ng/mL), cholera toxin (100 ng/mL), hydrocortisone (0.5 µg/mL), horse serum (5%), and penicillin/streptomycin (100 U/mL/100 µg/mL). The cells were incubated in an incubator with 5% carbon dioxide (CO_2_)/95% humidified air at 37 °C. The cells were sub-cultured upon ~80–90% confluency. All the assays were performed with experimental media containing 5% FBS, unless stated otherwise. The σ2RL SYA014 was dissolved in dimethyl sulfoxide (DMSO) and acetone at a 1:9 ratio. Cisplatin was dissolved in 0.9% sodium chloride (normal saline). Stocks of siramesine and PB28 were prepared in dimethyl sulfoxide (DMSO).

### 2.3. Cell Viability and Proliferation Assays

The cells suspended in 100 μL media were seeded at a density of 2 × 10^4^/well in 96 well plates and allowed to attach over 24 h in media containing 5% FBS. Cells were treated with SYA014 (0–200 μM) in 1 μL solvent. MDA-MB-231 cells were also treated with cisplatin (0–200 μM). The control cells were treated with the equivalent volume of the respective solvent (1:9 ratio DMSO/acetone for σ2RLs, and normal saline for cisplatin). MDA-MB-231 cells were treated for 24, 48, and 72 h. All other cells (HEK 293, MCF-10, and MDA-MB-468) were treated for 48 h. To determine the effect of viability of SYA014 on spheroids (3D) culture, MDA-MB-231 cells were seeded in 96-Well, U-shaped-bottom Nunclon Sphera plates (Thermo-Scientific, Waltham, MA) at 2.5 × 10^4^ cells per well in media supplemented with (5% FBS). The cells were incubated at 37 °C/5% CO_2_ and spheroids were formed after 24 h. The cells were then exposed to SYA014 (0–200 μM) and the treatment was repeated after 24 h for a total exposure of 48 h. Resazurin reagent (20 μL of 0.05%) was then added to each well and incubated for 1–2 h, after which the cell viability was determined as previously described [14,17]. The fluorescence was measured at 560 nm excitation wavelength and a detection wavelength of 590 nm using the FLx 800 Microplate Fluorescence Reader from BioTek (Winooski, VT. USA). To determine the effect of SYA014 on MDA-MB-231 and MDA-MB-468 cell proliferation, cells were plated at 20,000/well in 24-well plates for 24 h and treated with SYA014 (0–5 μM) for 48 h. Viable cells in triplicate wells were counted using the Countess II Automated Cell Counter (Life Technology Corporation (Grand Island, NY, USA)). The significance of the effect was established using GraphPad Prism version 5.0.

### 2.4. Colony Formation Assay

MDA-MB-231 and MDA-MB-468 cells were seeded at a density of 1.5 × 10^5^/mL in T-25 flasks and incubated overnight. The next day, cells were treated with SYA014 (0–20 μM) for 48 h. After the completion of incubation, cells were trypsinized, counted and re-plated in 6-well plates at a density of 500–1000 cells per well. Cells were maintained in complete growth media for 12 days. After the completion of the incubation period, cells were fixed with a solution of acetic acid in methanol (1:7) and stained with 1% crystal violet solution prepared in methanol. Colonies of MDA-MB-231 and MDA-MB-468 cells consisting of 50 or more cells were counted using the NIH ImageJ [18], http://rsb.info.nih.gov/ij/) accessed on 16 May 2018. The plating efficiency (PE) was determined by dividing the number of colonies formed by the number of cells seeded under untreated conditions × 100. The surviving fraction (SF) was calculated by dividing the PE of treated samples by the PE of control × 100 [19].

### 2.5. Spheroid Morphology Experiments

To evaluate the effect of SYA014 on MDA-MB-231 pre-formed spheroids, MDA-MB-231 cells were seeded at 5 × 10^3^ cells/well in a 96-Well, U-shaped-bottom Nunclon Sphera plate and incubated at 37 °C/5% CO_2_. Cells were incubated for 3 days for the formation of spheroids and then the pre-formed spheroids were treated with SYA014 (0–10 μM) for 48 h. Next, staining of the spheroids was performed with acridine orange/ethidium bromide (AO/EB) solution (5 μg/mL). Stained spheroids were subjected to fluorescence microscopy using Nikon Ti Eclipse Microscope and the images were captured at 4x magnification.

MDA-MB-231 cells were plated in 96-Well, U-shaped-bottom Nunclon Sphera plate at the density of 2 × 10^4^ cells/mL. Cells were then treated with varying concentrations of SYA014 (0–10 μM) and incubated at 37 °C/5% CO_2_ for a period of 24 h. The drug treatment was repeated after 24 h and the cells incubated for another 24 h. The effect of SYA014 on the inhibition of the formation of spheroids was evaluated after a total of 48 h exposure to SYA014. The size and morphology of the spheroids was continuously monitored for 11 days. Brightfield images were captured using a Nikon Ti Eclipse Microscope to monitor the progress of the inhibitory effect. A total of 50% of the media in each well was replaced with fresh media, and bright field images were obtained every day until 11th day. On the 11th day, the spheroids were stained with acridine orange/ethidium bromide (AO/EB) solution (5 μg/mL). Stained spheroids were examined by fluorescence microscopy using a Nikon Ti Eclipse Microscope, with the images captured at 4x magnification.

### 2.6. Cell Migration Assay

The potential of SYA014 to inhibit the migration and invasion of MDA-MB-231 cells was determined using the wound-healing/scratch technique. Ibidi culture inserts in ibidi µ-slide 8-well glass bottom chamber slides were used to generate scratch/wound with two confluent monolayers of cells separated by the cell-free zone. MDA-MB-231 cells were seeded at a density of 5 × 10^4^ cells in ibidi cell culture inserts, incubated to confluence, and serum-starved for 24 h. Cells were treated with SYA014 (0–10 μM) for 24 h and wound closure was monitored using a Nikon Eclipse Ti 100 inverted microscope (Nikon Instruments Inc., Melville, NY, USA). Wound closure images were captured at 0, 6, 12, and 24 h post-treatment and the images were analyzed as previously described [20]. The effect of SYA014 on the migration of MDA-MB-231 cells was calculated by counting the number of migrated cells to the cell-free area. Data were analyzed using GraphPad Prism version 5.0.

### 2.7. Cell Invasion Assay

The effect of SYA014 on cell invasion (2D) was determined using Matrigel-coated 24-well plates with 8 µm pore inserts. The 24-well plate with Matrigel invasion inserts (Corning, Bedford, MA) was rehydrated with serum-free DMEM with glutamate for 2 h at 37 °C/5% CO_2_. MDA-MB-231 cells (2 × 10^5^ cells/mL) suspended in 500 μL media containing 0.1% FBS SYA014 (0–10 μM) were added in the upper chamber of the inserts while 750 μL media with 10% FBS was placed into the lower chambers as a chemo-attractant. The plates were then incubated at 37 °C/5% CO^2^ for 22 h to allow the cells to invade from the upper chamber to the lower chamber of inserts through the Matrigel. The invading cells were rinsed with phosphate buffered saline (PBS), fixed with 4% formaldehyde in PBS, permeabilized with 100% methanol, and stained with 2% crystal violet for 1 h. The Olympus IX70 microscope was used to image the invading cells using the 4x objective. Invaded cells were counted using Imagej [18], http://rsb.info.nih.gov/ij/ accessed on 16 May 2018, and quantified using GraphPad Prism version 5.0.

The ability of SYA014 to inhibit invasion of cells from a formed spheroid in 3D format was determined using a protocol for this assay that was adapted as described by Vinci et al. [21]. Briefly, MDA-MB-231 cells were plated in 96-Well, U-shaped-bottom Nunclon Sphera plate (Thermo-Scientific, Waltham, MA, USA) at a density of 5 × 10^4^ cells/mL in complete growth media. Following the formation of compact spheroids, 850 μL of BD Matrigel (Corning, Bedford, MA, USA) was pipetted into the 10 pre-chilled micro-centrifuge tubes. SYA014 (3.4 μL–34 μL) were added to each tube to yield a final concentration of 0–20 μM of SYA014 in the mixture. The mixture was mixed gently and 100 μL of growth media from each well containing the spheroid was carefully replaced with 100 μL of Matrigel/SYA014 mixture. The plate was incubated for 1 h to embed the spheroids while solidifying the Matrigel, after which an additional 100 μL of Matrigel containing SYA014 was added to the corresponding wells (0–20 μM). The bright field images of Matrigel-embedded spheroids were taken at 0 h and after 72 h using a Nikon Ti Eclipse Microscope at 4x magnification. The area of invasion was quantified using NIS Element software, and the data analyzed using GraphPad Prism version 5.0.

### 2.8. Apoptosis Assays

The effect of SYA014 on the induction of cell death was determined using three different assays. We employed the modified ethidium bromide/acridine orange (EB/AO) staining method [22] and the Annexin V/propidium iodide staining method followed by flow cytometry analysis and caspase activation analysis using Caspase-Glo^®^ assay system.

#### 2.8.1. Ethidium Bromide/Acridine Orange (EB/AO) Staining

MDA-MB-231 and MDA-MB-468 cells were seeded at a density of 5 × 10^4^ cells/mL into a 96-well plate and incubated overnight at 37 °C in 5% CO_2_/95% humidified air. The next day, cells were treated with SYA014 (0.1–20 μM) and incubated. After 24 h of incubation with SYA014, the drug treatment was repeated and incubated for another 24 h. After 48 h of the first treatment, 10 μL of EB/AO solution (EB and AO solutions were prepared at 100 μg/mL for each separately in DPBS, and 1 part of EB and 1 part of AO from each of the 100 μg/mL solutions were mixed to prepare the EB/AO solution) was added directly into each well containing media and incubated for 10 min in the dark. The stained cells were observed under a Nikon Eclipse Ti 100 inverted fluorescent microscope (Nikon Instruments, Inc., Melville, NY, USA). Changes in the morphology of the nuclei were observed and images were captured for the control and treated cells.

#### 2.8.2. Annexin V/Propidium Iodide (AV/PI) Staining and Flow Cytometry Analysis

Next, we performed Annexin V/Propidium iodide (AV/PI) staining and flow cytometry analysis to determine the cell death mechanism induced by SYA014. Annexin V is a phospholipid-binding protein with a high affinity for phosphatidylserines, which are normally located in the inner part of the cell membrane but translocate to the surface during early apoptosis due to loss of membrane integrity. The propidium iodide, on the other hand, can enter the cells only when the cell membrane is compromised as seen in late apoptosis as well as in necrosis. MDA-MB-231 and MDA-MB-468 cells (5 × 10^5^ cells/well) suspended in media supplemented with 5% FBS were plated in 6-well culture plates (Costar Corning, NY, USA) and incubated at 37 °C in 5% CO_2_/95% humidified air and allowed to attach overnight. The next day, the cells were treated with SYA014 (0–10 μM) continuously for 48 h. The mode of cell death was determined as per the manufacturer’s protocol using the ApopNexin fluorescein isothiocyanate (FITC) apoptosis detection kit (EMD Millipore, Temecula, CA, USA). Briefly, after the completion of incubation, the cells were washed with DPBS and harvested with accutase. The cells were pelleted using a centrifuge at 500× *g* for 5 min. The cell pellets were washed with 1X DPBS (2 × 5 mL) on ice and re-suspended in 1X binding buffer. Annexin V FITC and PI were added to the cell suspensions and incubated for 10 min at room temperature in the dark. Flow cytometry analysis was performed using the FACS Calibur with Cell Quest Pro software (BD Biosciences, Franklin Lakes, New Jersey, USA).

#### 2.8.3. Caspase Activation Assay

The effect of SYA014 on the activation of Caspases in MDA-MB-231 cells was evaluated using the Caspase-Glo 3/7 and caspase-Glo 8 and 9 Assay Kits (Promega, Madison, WI) according to the manufacturer’s instructions. Briefly, MDA-MB-231 cells were plated in white-walled 96-well plates at a density of 1.5 × 10^4^ cells/well (for caspase 3/7) and 3 × 10^4^ cells/well (for caspase 8 and 9) in 100 μL media with 5% FBS. The cells were incubated at 37 °C in 5% CO_2_/95% humidified air overnight. The next day, cells were treated with SYA014 (0–5 μM) continuously for 48 h. After the completion of the incubation, the multiplexing reagent (prepared by mixing Caspase-Glo^®^ Buffer and the lyophilized substrate) was added to each well at 100 μL/well while protecting it from light. The cells were incubated at room temperature for 1 h. The luminescence readings were measured using a luminometer. The relative maximum luminescence intensity was quantified using the GraphPad Prism version 5.0.

### 2.9. Cell Cycle Analysis

The effect of SYA014 on the cell cycle was determined in MDA-MB-231 cells using the propidium iodide staining protocol. Briefly, MDA-MB-231 and MDA-MB-468 cells were plated at a density of 2 × 10^5^ cells per well in 6-well plates in media supplemented with 10% FBS. Cells were incubated at 37 °C in 5% CO_2_/95% humidified air overnight for attachment. The next day, media was replaced with fresh media containing 5% FBS and then cells were treated with SYA014 (0–10 μM) and incubated for 24 h. SYA014 treatment was repeated after 24 h and incubated for another 24 h. After 48 h of incubation from the first treatment, cells were harvested by scraping, washed with PBS, and centrifuged at 300× *g* for 5 min. The cell pellets were re-suspended in 100 μL PBS and passed through a 28 gauge 5/8 inch needle to keep it as a single cell suspension. Cells were fixed with the dropwise addition of cold 70% ethanol while vortexing, and cells were stored at −20 °C overnight. The next day, cells were washed with PBS (2 times), and centrifuged at 500× *g* for 5 min. Modified Vindelov’s reagent (ribonuclease A and propidium iodide in PBS) [23] was added to the cells and the phase distribution of cells was determined using a Becton Dickinson FACSort flow cytometer with CellQuest software (Mansfield, MA, USA). The percentage of cells in each phase was determined in the gated population of singlet cells and plotted using the GraphPad Prism version 5.0.

### 2.10. Western Blot Analysis

The effect of SYA014 on the expression of other apoptotic, anti-apoptotic (survival), death receptor, cell cycle, and angiogenesis proteins in MDA-MB-231 cells after treatment was evaluated by western blot electrophoresis. Cells were plated in 6-well tissue culture plates at the density of 2 × 10^5^ cells/well in media supplemented with 10% FBS and allowed to adhere to the plates overnight at 37 °C in 5% CO_2_/95% humidified air. Prior to treatment, the complete media was replaced with 5% supplemented media and treated with SYA014 (0–10 μM). The treatment was repeated after 24 h for a total time of exposure of 48 h. Briefly, the cells were washed with PBS, and lysed with radioimmunoprecipitation assay (RIPA) buffer supplemented with 1X protease inhibitors cocktail (Sigma, St. Louis, MO, USA). The protein concentration in the lysates was evaluated using a Pierce BCA protein assay kit (Thermo Scientific, Rockford, IL, USA). Lysates containing 50 μg of protein were mixed with Laemmli sample buffer, and placed in a boiling water bath for 5 min. Aliquots of each sample were resolved using a 10–20% gradient on an SDS-PAGE gel. Resolved proteins were transferred onto a polyvinylidene difluoride (PVDF) membrane. The membranes were blocked with 5% fat-free milk (Sigma, St. Louis, MO, USA) for 1 h at room temperature, then immunoblotted using primary antibodies against different protein groups including angiogenesis, death receptors, apoptosis, and cell cycle purchased from Cell Signaling Technology (Danvers, MA, USA) and incubated overnight at 4 °C. The next day, membranes were incubated with horseradish peroxidase-linked anti-rabbit IgG secondary antibodies from Santa-Cruz Biotechnology (Santa Cruz, CA, USA) for 90 min. Enhanced chemiluminescent (ECL) substrate was used for detection of horseradish peroxidase (HRP) activity from antibodies, and the chemiluminescent images were obtained using a ChemiDoc imaging system (Bio-Rad, Hercules, CA, USA).

### 2.11. σ2. R Blockade Experiment with RHM-1

σ2R Blockade was performed according to the method reported previously with slight modifications [24]. The cells were seeded at a density of 1 × 10^4^ cells/well in 96 well plates and allowed to attach over 24 h in complete media. Cells were initially treated with RHM-1 at 10 µM, followed by SYA014 at 25 µM, PB28 at 25 µM and siramesine at 10 µM, either alone or with RHM-1 for 24 h. The control cells were treated with an equivalent volume of the respective solvent. Resazurin reagent (20 µL of 0.05%) was then added to each well and incubated for 1–2 h. The fluorescence was measured at a 560 nm excitation wavelength and a detection wavelength of 590 nm using the FLx 800 microplate fluorescence reader from BioTek (Winooski, VT, USA) to determine the cell viability. Results were plotted using GraphPad Prism version 5.0 software (San Diego, CA. USA). The significance of effect was established using GraphPad Prism version 5.0.

### 2.12. Cell Viability Study for the Combination Effect of SYA014 and Cisplatin

Cell viability studies were performed to evaluate the cytotoxicity of cisplatin in MDA-MB-231, MDA-MB-468, and MCF10A cells, and also the combination effect of SYA014 with cisplatin in the above-mentioned cell lines. The cells were seeded at a density of 5000 cells/well in 96 well plates and allowed to attach over 24 h in complete media. For the 72 h cytotoxicity studies of cisplatin (0.1–50 µM), 5% FBS containing media was used for MDA-MB-231 and MDA-MB-468 cells, while assay media was used for MCF10A cells. For the combination studies, cells were initially treated with cisplatin at 0.5 µM followed by SYA014 (2–20 µM) for 72 h in the same media as the corresponding cytotoxicity studies. The control cells received equivalent volume of the respective solvent (1:9 ratio DMSO/acetone for SYA014, and normal saline for cisplatin). Cell viability was measured as in Section 2.3. Results were plotted using GraphPad Prism version 5.0 software (San Diego, CA, USA). The significance of effect was established using GraphPad Prism version 5.0.

### 2.13. Statistical Analysis

The statistical analysis of data was performed using GraphPad Prism version 5.0 for Windows (San Diego, CA, USA). Data results were expressed as the mean ± standard error of the mean (SEM). Data were also analyzed using one-way analysis of variance (ANOVA). Statistical differences between control and treated groups were determined by Dunnett’s post-test comparisons or Tukey’s multiple comparison test. Significance was defined as * *p* < 0.05; ** *p* < 0.01; and *** *p* < 0.001.

## 3. Results

### 3.1. SYA014 Inhibits Cell Proliferation in TNBC Cell Lines

Cell proliferation without restraint is an important aspect of cancer progression. The effect of SYA014 against cell proliferation was evaluated in two TNBC cell lines, MDA-MB-231 and MDA-MB-468. Cells were treated with SYA014 at different concentrations (0.5, 1, 2 and 5 μM) for 48 h. SYA014 showed a decrease in cell proliferation at 0.5 μM and a halt at 1 μM in MDA-MB-231 cells, while a gradual effect of SYA014 on the cell proliferation was observed in MDA-MB-468 cells (Figure 2A).

### 3.2. SYA014 Reduces the Colony Formation Ability of TNBC Cell Lines

Clonogenic or colony formation assays were conducted with MDA-MB-231 and MDA-MB-468 cells to determine the ability of SYA014 to inhibit individual cells from forming colonies. Cells treated with different concentrations of SYA014 (0–10 μM for MDA-MB-231 and 0–20 μM for MDA-MB-468 cells) showed dose-dependent reduction in the colony formation for both cell lines. SYA014 effectively inhibited colony formation at 10 and 20 μM in MDA-MB-231 and MDA-MB-468 cells, respectively, as shown in Figure 2B. The cell survival fractions (Figure 2C) were derived from Figure 2B, which shows a decrease in the formation of colonies in MDA-MB-231 and MDA-MB-468 cells treated with SYA014. In MDA-MB-231 cells, formation of colonies was reduced by 25% and 65% at 5 μM and 10 μM of SYA014, respectively, while the reduction was 66%, 86% and 90% at 5, 10 and 20 μM concentrations of SYA014, respectively, in MDA-MB-468 cells (Figure 2C).

### 3.3. Effect of SYA014 on MDA-MB-231 Spheroids

Although cellular assays using 2D monolayers are standard for the initial screening of anticancer drugs, they do not mimic the true 3D characteristics of a tumor microenvironment that may impose drug penetration constraints. In fact, spheroid models are better at replicating features such as nutrient and oxygen gradients which vary from the external proliferating zone through the internal quiescent zone to the necrotic core of the tumor. However, 3D spheroid models lack certain important elements of in vivo solid tumors, such as multicellular heterogenicity, blood vessels and complex internal gradients of signaling factors. Nevertheless, a 3D spheroid model-based drug screening provides more useful information than 2D monolayer assays.

#### 3.3.1. SYA014 Disintegrates Pre-Formed Spheroids

Initially, MDA-MB-231 cells were seeded at 2.5 *×* 10^4^ cells per well in U-shaped bottom microplates for 24 h to form spheroids. The spheroids were treated with SYA014 at a concentration range of 0–200 µM. Repeat treatments were made after 24 h, followed by further incubation for another 24 h for a total exposure of 48 h. Cell viability assays revealed a concentration-dependent effect of SYA014 on MDA-MB-231 spheroid cells with an IC_50_ of 17 µM (Figure 3A). Further, to observe the disintegration effect of SYA014 on pre-formed spheroids, MDA-MB-231 cells were seeded in U-shaped bottom microplates to form 3D spheroids and treated with SYA014 at 1, 2 and 5 µM for 48 h followed by acridine orange/ethidium bromide (AO/EB) staining. As shown in Figure 3B, SYA014 induced concentration-dependent degeneration of the pre-formed spheroids with maximal effect at 5 µM.

#### 3.3.2. SYA014 Blocks Spheroid Formation

One of the important advantages of cells growing in 3D spheroids compared to 2D or monolayer cultures is that spheroids tend to form a tissue-like environment which can differentiate and exhibit different characteristics of solid tumor phenotype. We further evaluated the effect of SYA014 on preventing MDA-MB-231 spheroid formation for up to 11 days. Cells treated with SYA014 at 1–10 µM blocked the spheroid formation in a dose-dependent manner (Figure 3C).

### 3.4. SYA014 Inhibits MDA-MB-231 Cell Migration and Invasion

Cancer metastasis is an advanced stage of malignancy which involves multi-step processes in spreading the cancer cells into the surrounding healthy tissues and the vasculature and then to distant organs of the body system. Migration and invasion are the two important aspects of the metastasis cascade. TNBC is more aggressive and highly invasive in nature. In this context, we have evaluated the ability of SYA014 to prevent the migration and invasion of MDA-MB-231 cells using cellular assays.

To evaluate the effect of SYA014 at preventing the migratory nature of MDA-MB-231 cells, the migration assay was performed. Cells treated with SYA014 at two different time points, i.e., 12 h and 24 h, showed significant reductions in the movement of cells towards the wound area as compared to the control (Figure 4A,B). SYA014 effectively restricted the migration of cells within 12 h of treatment (Figure 4A). It prevented MDA-MB-231 cell migration by 40%, 61%, and 72% at 2, 5, 10 µM, respectively, as compared to the controls at 24 h (Figure 4B).

The invasive capacity of the tumor cells into surrounding normal tissues is another important aspect of cancer metastasis leading to high mortality. Importantly, TNBC is known to be highly invasive and the evaluation of drugs’ effects on the inhibition of the invasion of cells using different invasion assays provides important information on the migratory behaviors of tumor cells. It also allows us to understand the underlying molecular mechanisms for developing novel strategies in cancer drug discovery and development. To determine the anti-invasion capacity of SYA014, monolayer MDA-MB-231 cells were treated with SYA014 at 2, 5, 10, and 20 µM concentrations in Transwell Matrigel invasion chambers for 22 h. The invasion of cells towards complete media containing 10% FBS, which serves as a chemo-attractant, was evaluated with crystal violet staining followed by microscopic imaging. The results show that SYA014 prevents the invasion of MDA-MB-231 cells in a concentration-dependent manner (Figure 4C). Quantification of the images by counting the number of cells using ImageJ software revealed decreasing numbers of invaded cells by 24%, 36%, 51%, and 85% compared to the control group (SYA14 at 0 µM) for SYA014 at 2, 5, 10, and 20 µM concentrations, respectively (Figure 4D). To further understand the anti-invasive effects of SYA014, a 3D invasion assay using MDA-MB-231 spheroids (3D) was performed. The ability of SYA014 to prevent the spheroids from invading other tissues was evaluated. As depicted in Figure 4E,F, the area of invasion through the Matrigel was significantly decreased compared to the control after 72 h of SYA014 treatment. SYA014 at 2, 5, 10, and 20 µM showed a concentration-dependent reduction in the area of invasion compared to control in the 3D spheroid invasion assay (Figure 4F).

### 3.5. Mode of Cell Death

#### 3.5.1. SYA014 Induces DNA Fragmentation in TNBC Cells

Apoptosis is characterized by a series of morphological and biochemical changes occurring in the tumor cells. Internucleosomal fragmentation of genomic DNA is one of the key events in apoptotic cell death. Thus, SYA014 was evaluated for its apoptosis induction through DNA fragmentation in TNBC cells (MDA-MB-231 and MDA-MB-468 cells) (Figure 5). Cells treated with SYA014 at different concentrations for 48 h, followed by AO/EB staining and fluorescence microscopy imaging, revealed apoptotic cell death as shown in Figure 5A,B for MDA-MB-231 and MDA-MB-468 cells, respectively. In MDA-MB-231 cells, SYA014 at 10 µM showed high levels of red-stained nuclei indicating SYA014 induced DNA fragmentation (Figure 5A).

#### 3.5.2. SYA014 Induces Apoptosis in TNBC Cells

Next, we evaluated the effects of SYA014 on apoptosis using Annexin V/FITC PI staining in MDA-MB-231 and MDA-MB-468 cells (Figure 6A,B). Cells treated with 0−20 μM SYA014 exhibited a concentration-dependent apoptotic effect in both TNBC cell types. MDA-MB-231 cells treated with SYA014 showed increasing populations of early apoptotic cells with increasing SYA014 concentrations. A higher percentage of late apoptotic cells, 89.3%, 86.5% and 68.7%, was observed at 5, 10 and 20 μM SYA014 concentrations, respectively (Figure 6A,C), while MDA-MB-468 cells treated with SYA014 showed increasing levels of early apoptotic cells in a concentration-dependent manner with high percentages at 5, 10 and 20 μM concentrations of SYA014 (Figure 6B,D).

#### 3.5.3. SYA014 Arrests Cell Cycle in TNBC Cells

Cell cycle comprises different stages of cell division involving various changes occurring at the cellular level, including the amount of DNA. This is a continuous process, and any abnormal condition disrupts the phases of cell division. Cell cycle analysis shows the population of cells based on the varying DNA content within the cells as indicative of a normal or a disrupted cell cycle. Thus, we performed the cell cycle analysis of MDA-MB-231 and MDA-MB-468 cells treated with SYA014 for 48 h and analyzed the resulting data with PI staining and flow cytometry (Figure 7). The results reveal that SYA014 arrests the cell cycle at the G0/G1 phase in MDA-MB-231 cells at 5 and 10 µM (Figure 7A,C), and in MDA-MB-468 cells at 2 and 5 µM (Figure 7B,D), respectively.

#### 3.5.4. SYA014 Induces Apoptosis via Caspase 9 and 3/7 Activation

Caspases belong to a diverse clan of the cysteine protease family, which serves as the key components of the machinery responsible for apoptosis. An extrinsic or intrinsic death signal generally activates the apoptotic caspases. Reports indicate that apoptotic initiator caspases (caspase 8 and 9), along with apoptotic effector caspases (caspase 3/7), play significant roles in promoting the ordered disassembly of the cellular proteins involved in signal transduction pathways. Therefore, caspase activation in MDA-MB-231 cells upon treatment with SYA014 was measured using a caspase GLO assay kit (Promega, Madison, WI). Results show that SYA014 activates caspase 9 by about 30% at 1 and 2 µM and caspase 3/7 by about 300% at 2 and 5 µM in MDA-MB-231 cells (Figure 8).

### 3.6. SYA014 Operates through Intrinsic Apoptotic Mechanism

The results from the cell proliferation, cell migration, invasion and spheroid inhibition assays shown in the above sections reveal the antiproliferative potential of SYA014 in two representative TNBC cells, MDA-MB-231 and MDA-MB-468. The apoptosis and caspase analysis support the significant antiproliferative effects of SYA014 in both TNBC cell lines. Next, we wanted to probe the antiproliferative mechanism of SYA014. To this effect, the important molecular events associated with apoptotic cell death were probed by determining the levels of key proteins involved in various pro- and anti-apoptotic signaling cascades, and the death receptors and cell cycle proteins were analyzed as well. To achieve this, MDA-MB-231 cells were treated with 0–10 µM SYA014 for 48 h, followed by western blot analysis of the cell lysates (Figure 9). To begin with, analysis of the pro-apoptotic proteins, Caspase, PARP, Bax, Bak and Cyt C, showed decreased levels of caspase 3 and PARP as compared to control groups, indicating the involvement of the intrinsic apoptotic pathway (Figure 9A,B). Next, western blot analysis of pro-survival proteins such as Bcl-2, Bcl-XL and Mcl-1 showed significant concentration-dependent decreases in the amount of the anti-apoptotic Bcl-family proteins in cells treated with SYA014 (Figure 9C,D). Similar trends were observed in the case of death receptor proteins where DR5, TNF-R1 and RIP were found to decrease as the concentration of SYA014 increased (Figure 9E,F). However, SYA014 did not show any significant changes in the expression of proteins involved in cell cycle regulation (Figure 9G,H).

### 3.7. Cytotoxic Effects of SYA014 Are Independent of the σ2R

Sigma-2 receptors are 8–10 times more abundant in proliferating solid tumors such as TNBC than quiescent cells, hence they are proposed as biomarkers of proliferating cells in solid tumors. Studies have demonstrated that several σ2R ligands kill tumor cells by apoptosis and by various signaling pathways that include the p53 apoptotic pathway, caspase-dependent apoptotic pathway, impaired cell cycle progression, mTOR pathway, and EGFR signaling pathway. As previously reported, the binding affinities (*K_i_* values) towards σ1 and σ2 receptors of SYA014 are 5.1 and 3.5 nM with a (*K_i_* σ1/*K_i_* σ2) ratio of 1.5 [14]. However, there is no strong evidence to establish the role of σ2 receptors in the antiproliferative effects of the σ2 ligands. Therefore, we explored the role of σ2 receptors in inducing cytotoxic effects of SYA014 in TNBC cell lines, MDA-MB-231 and MDA-MB-468 using σ2 receptor blockade. To accomplish this, cells were treated with 25 μM of SYA014 in the presence and absence of RHM-1 for 24 h. Since the IC_50_ values for SYA014 in MDA-MB-231 and MDA-MB-468 cells are 8 and 4 μM (48 h), respectively [14], we used SYA014 at a concentration of 25 μM (24 h) to allow a larger window of observation for its cytotoxic effect when SYA014 is treated in presence of RHM-1. Siramesine and PB28 were used as positive controls. Since RHM-1 is a non-toxic (>100 μM) sigma-2 antagonist [16], by incubating TNBC cells with SYA014 in the presence of RHM-1, we wanted to know whether the antiproliferative effects of SYA014 are σ2R dependent or independent. Data obtained from cell viability assays in MDA-MB-231 and MDA-MB-468 cells show high cytotoxicity in the cells treated with SYA014 and RHM-1 or SYA014 alone (Figure 10). σ2 receptor saturation with RHM-1 did not alter the cytotoxic effects of SYA014 in both TNBC cells. Interestingly, Siramesine and PB28 also showed a similar profile (Figure 10).

### 3.8. Cytotoxic Effects of Combination Treatments with Cisplatin and SYA014

To determine the optimal concentration of cisplatin for the combination treatments, cytotoxicity assays were conducted with cisplatin at 0–50 µM concentrations against MDA-MB-231 and MDA-MB-468 cells along with MCF10A cells for 72 h (Appendix A). The results show that cells treated with cisplatin in the concentration range of 1–50 µM have less than 50% cell viability (Appendix A). Cisplatin at 0.1 µM showed no cytotoxic effects in MDA-MB-231 cells, while cytotoxicity was found to be about ~10 and ~15% for MDA-MB-468 and MCF10A cells at 0.1 µM cisplatin, respectively. However, cisplatin at 0.5 µM showed ~10, ~20 and ~35% cell death in MDA-MB-231, MDA-MB-468 and MCF10A cells, respectively (Appendix A). Thus, in order to reduce the cisplatin mediated toxicity to normal cells but maintain anticancer effects on tumor cells, a concentration of 0.5 µM cisplatin was selected from the test range of 0.1 to 50 µM cisplatin for the combination treatments with SYA014. The concentration of cisplatin to test was based on the assumption that, when 0.5 µM cisplatin is co-treated with SYA014, it would provide an opportunity to monitor the cytotoxicity window on TNBC cell lines in the range of 0–50%.

Most of the current chemotherapeutic agents are highly effective at treating various types of cancer malignancies. However, there are several drawbacks with the use of standard agents, such as drug resistance and toxicity to normal tissues. The development of novel anticancer agents is one of the alternative strategies for the effective treatment of various types of cancers. These novel agents can be used as monotherapy or in combination with standard drugs in use. A combination of drugs is one of the widely used strategies for treating cancer to mitigate toxicity side effects. Such strategies could be more effective than monotherapies owing to their multi-directional and/or multi-pathway mechanism of action on tumor cells. The combination therapy could also impart a synergistic effect and enhance the overall impact on cancer tissues, while reducing the standard drug-induced toxicity to normal cells. Therefore, we evaluated the combination effect of the widely used anticancer drug cisplatin along with SYA014 in TNBC cells (MDA-MB-231 and MDA-MB-468) and MCF10A (normal breast epithelial cells). For this, TNBC and MCF10A cells were co-treated with 0.5 µM cisplatin and SYA014 (0–20 µM) for 72 h (Figure 11). In MDA-MB-231 cells, cisplatin alone showed ~20% cytotoxicity and SYA014 at 20 µM showed ~40% cell death. However, SYA014 at remaining concentrations, i.e., 2, 5 and 10 µM, did not show any cytotoxicity (Figure 11A). Interestingly, a combination of cisplatin with SYA014 at 20 µM enhanced the cytotoxic effects, with ~80% cell death as compared to ~40% for 20 µM SYA014 alone. Similarly, SYA014 at 5 and 10 µM in combination with cisplatin showed ~25% and ~35% cell death, while SYA014 alone at the same concentration was not cytotoxic (Figure 11A). A similar pattern was observed in MDA-MB-468 cells, where cisplatin alone showed about ~15% cytotoxicity and SYA014 alone at 20, 10 and 5 µM showed ~85%, ~35% and 10% cell death, respectively, while at 2 µM no cytotoxicity was observed (Figure 11B). When SYA014 was co-treated with cisplatin, SYA014 at 20 µM exhibited almost 100% cytotoxicity, whereas SYA014 at 10, 5 and 2 µM showed ~45%, ~30% and 20% cell death, respectively (Figure 11B). No significant cytotoxicity difference was observed for SYA014 at 20 µM when MDA-MB-468 cells were treated either alone or in combination with cisplatin. However, this difference was significant when cisplatin was co-treated with SYA014 at 10, 5 and 2 µM, as compared to SYA014 treatment alone at their respective concentrations. Importantly, MCF10A cells treated with cisplatin showed ~40% toxicity and SYA014 at 20 µM showed ~50% cytotoxicity. However, SYA014 was found to be least effective at killing MCF10A cells at other concentrations (Figure 11C). It is also interesting to note that cells treated with SYA014 in combination with cisplatin did not show any additional toxic effects on MCF10A cells except for SYA014 at 20 µM, which showed ~70% cytotoxicity as compared to ~50% cell death for SYA014 alone at 20 µM (Figure 11C). As shown in the Figure 11C, no significant difference in toxicity was observed for cisplatin (cytotoxicity ~40%) and the combination treatment of cisplatin with SYA014 at 10, 5 and 2 µM (cytotoxicity range 40%–50%). The cytotoxicity observed in MCF10A cells for cisplatin and SYA014 co-treatment at 10, 5 and 2 µM can be attributed mostly to the cytotoxic effect of cisplatin alone.

## 4. Discussion

Triple negative breast cancer (TNBC) is a type of breast cancer which lacks all three standard molecular markers, i.e., estrogen receptors (ER), progesterone receptors (PR), and human epidermal growth factor receptor type 2 (HER2) expression. TNBC is known to have a higher recurrence rate than other breast cancer subtypes. Due to the lack of these specific receptors, there are limited treatment options for TNBC. Targeted therapies for other subtypes of breast cancer are highly effective with improved survival rate; however, the same is not true for TNBC. Chemotherapy is the only treatment option for TNBC with several limitations such as high relapse rate and toxicity to normal cells. Therefore, there is a dire need for the development of novel therapeutic modalities with no or minimal side effects for the treatment of TNBC. In this regard, we have developed two compounds (SYA013 and SYA014) based on the haloperidol core structure which showed potential anticancer properties. In our previous work, we have reported on the cytotoxicity potentials of SYA013 and SYA014 against two TNBC model cell lines, MDA-MB-231 and MDA-MB-468. We have also demonstrated that these compounds show minimal cytotoxicity against MCF-10A cells, a nontumorigenic breast cell line. Based on those original findings, we further probed the anticancer properties of SYA013 in one of the representative TNBC cells, MDA-MB-231, and made an attempt to decipher the mechanism involved in the antiproliferative property of SYA013 [15]. In a continuation of these studies, we have now evaluated the anticancer effects of SYA014, an oxime analog of SYA013, against TNBC cells (MDA-MB-231 and MDA-MB-468) and possible mechanisms involved in its apoptotic effects. Initial studies demonstrated that SYA014 inhibits cell proliferation and colony formation in MDA-MB-231 and MDA-MB-468 cells in a concentration-dependent manner, indicating the antiproliferative effects of SYA014 against TNBC cells. SYA014 also prevented MDA-MB-231 spheroid formation and induced disintegration of pre-formed spheroids, as revealed by the cytotoxicity assay and acridine orange/ethidium bromide (AO/EB) staining. Thus, cell proliferation, colony formation assays, and studies with the inhibition of spheroid formation and disintegration of pre-formed spheroids demonstrate the suppressive effects of SYA014. In addition, SYA014 significantly reduced the migration and inhibited the invasion of MDA-MB-231 cells, as shown by the cell migration assay, 2D single cell, and 3D spheroid invasion assays. Results from these studies indicate the anti-invasive effects of SYA014. The metastatic growth of TNBC to distant organs in the body poses a significant clinical challenge. Further, the aggressive and highly invasive nature of TNBC limits treatment options [25]. Therefore, compounds such as SYA014, with antiproliferative and anti-invasive properties, may have potential for controlling cancer spread.

The fragmentation of nuclear DNA into oligonucleosomal size fragments is one of the key biochemical hallmarks of the late-stage apoptosis [26]. We have also shown that SYA014 induces DNA fragmentation in MDA-MB-231 and MDA-MB-468 cells, as demonstrated by AO/EB staining results. The effect of SYA014 in triggering apoptosis in TNBC cells was confirmed by the propidium iodide/annexin V-FITC apoptosis assay followed by flow cytometric analyses. Cell cycle analysis showed that SYA014 arrests the cell cycle at the G0/G1 phase in both TNBC cell lines. To illustrate the apoptosis-inducing effect of SYA014, we evaluated the mode of cell death by measuring the activation of caspases, namely caspase 8, 9 and, 3/7. Activation of caspases is triggered by the intrinsic and extrinsic signaling cascade, inducing the fragmentation of genomic DNA, thus leading to apoptotic cell death [27]. We observed a significant upregulation in caspase 3/7 (an apoptosis effector caspase) and a moderate increase in caspase 9 (an apoptosis initiator caspase). Caspase-9 is an essential initiator caspase for the intrinsic apoptotic pathway [28]. The decreased levels of procaspase-9 and down-regulation of procaspase-3, an executor caspase, indicates the triggering of the intrinsic apoptotic mechanism in TNBC cells. PARP also functions as an important marker for apoptosis [29]. The cleavage of PARP, a substrate of active caspase-3 from a 116 kDa band to a 89 kDa fragment, further supported the activation of caspase-3 [30]. The intrinsic apoptotic mechanism was further confirmed by western blot analysis with upregulation of other pro-apoptotic proteins, Bax and, Bak expression. In addition, anti-apoptotic family proteins such as B-cell lymphoma 2 (Bcl-2), myeloid cell leukemia 1 (Mcl-1), and Bcl-2 like protein X (Bcl-XL) expression was decreased. The evaluation of death receptor proteins, DR5, TNF-R1 and RIP and cell cycle regulator proteins, p21, p27, Cyclin D1, Cdk4 and Cdk4 did not show any significant effects, suggesting that the apoptosis induced by SYA014 was not by extrinsic pathway. Thus, the present data indicate the possibility that SYA014 induced the intrinsic mechanism involved in the apoptotic pathway of TNBC cell death. Siramesine, a well-known σ2R, follows the intrinsic apoptotic pathway and induces cell death by oxidative stress and mitochondria destabilization [31]. Another σ2R agonist and σ1R antagonist, PB28, induces caspase-independent apoptosis [32]. Other σ2R ligands such as WC-26, SV119, and RHM-138, with cytotoxic effects in different cancer cells, induce cell death by multiple signaling pathways [33]. Based on the results obtained in our previous study with SYA013 in MDA-MB-231 cells, we have outlined a possible apoptotic pathway as intrinsic in nature [15] and SYA014 seems to follow the same mechanism of action in inducing apoptosis.

There are several σ2R binding ligands (agonists/antagonists), such as siramesine, PB28, SV119 and WC-26, which showed significant in vitro antiproliferative effects [34,35]. However, it is still not clear whether σ2R binding of these ligands has any role in imparting/enhancing their anticancer properties. Recently, an interesting study involved widely known σ1 and σ2 ligands being screened in a series of cell lines, and the results revealed that there is no correlation between σ2R expression and the binding of ligands to their receptors, and observed antiproliferative properties with a few exceptions [36]. Our results with SYA014 in σ2R blockade are consistent with these earlier findings, where blocking of σ2R in MDA-MB-231 and MDA-MB-468 cells with the RHM-1 ligand had no significant impact on the cytotoxic effect of SYA014. Furthermore, siramesine and PB28 exhibited similar cytotoxicities, either alone or in the presence of the σ2R blocking ligand, RHM-1. Thus, it is evident from the results that the σ2R is unlikely to play a significant role in the anticancer effects of SYA014 in the representative TNBC cells.

Combination therapy is a therapeutic modality for the effective treatment of cancers. The use of two or more drugs certainly enhances the therapeutic efficacy compared to monotherapy, due to several factors such as targeting multiple signaling pathways in tumor cells, additive/synergistic effect and reducing a single drug overdose while addressing the multidrug resistance. Thus, the use of combination therapy offers multiple advantages over monotherapy. In particular, novel agents with different therapeutic targets provide an opportunity for developing new combination therapies. SYA014 exhibited significant antiproliferative effect in representative TNBC cell types and has the potential for exploration as a monotherapeutic agent. In addition, different doses of SYA014 (20, 10, 5 and 2 µM) in combination with cisplatin (0.5 µM) enhanced the cytotoxic effects in MDA-MB231 cells. Similar effects were observed in MDA-MB-468 cells with combination treatment of cisplatin with SYA014 at 10, 5 and 2 µM concentrations. Interestingly, co-treatment of cisplatin with SYA014 at 10, 5 and 2 µM concentrations in normal breast epithelial tissue cells (MCF10A) did not exhibit significantly increased cytotoxicity as compared to cisplatin treatment alone. Therefore, it is evident that the combination of cisplatin with SYA014 at 10, 5 and 2 µM concentrations did not increase toxicity to normal cells compared to cisplatin alone. Thus, SYA014 is effective in killing tumor cell types either alone or in combination with standard chemotherapeutic drugs.

## 5. Conclusions

This study and evaluation of SYA014 for the antiproliferative properties against TNBC cells reveals that SYA014 exhibits significant cytotoxic effects in two important TNBC cell lines through various inhibitory mechanisms. SYA014 successfully inhibits cell proliferation, colony formation in MDA-MB-231 and MDA-MB-468 cells, and inhibits spheroid formation, invasion and migration in MDA-MB-231 cells. Apoptosis, caspase activation and western blot analyses of key pathway proteins show that SYA014 induces cell death by the intrinsic apoptotic mechanism. Thus, SYA014 exhibits potent anticancer effects by disrupting important cellular events in TNBC cell lines, MDA-MB-231 and MDA-MB-468. This investigation also reveals that, despite high binding affinity and a moderate σ2 receptor selectivity of SYA014, the receptor does not appear to have a significant role in inducing the antiproliferative effects of SYA014 in TNBC cells. Nevertheless, SYA014 shows possible synergistic cytotoxicity in combination with cisplatin in both TNBC cells (MDA-MB-231 and MDA-MB-468). Our preliminary results of combination treatment with SYA014 and cisplatin in TNBC cell lines provide a possible direction in which to explore the synergistic effects of standard chemotherapeutic drugs with novel agents to reduce the toxicity issues caused by standard agents. However, further investigations are warranted for determining the effective range of concentrations to obtain high synergistic effects of combination treatments under various in vitro and in vivo settings, and for the development of novel combination treatment strategies against TNBC.

## Figures and Tables

**Figure 1 cancers-14-06047-f001:**
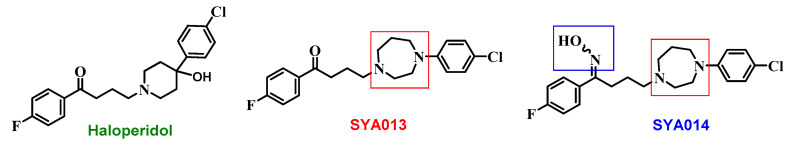
Chemical structures of haloperidol, SYA013, and SYA014.

**Figure 2 cancers-14-06047-f002:**
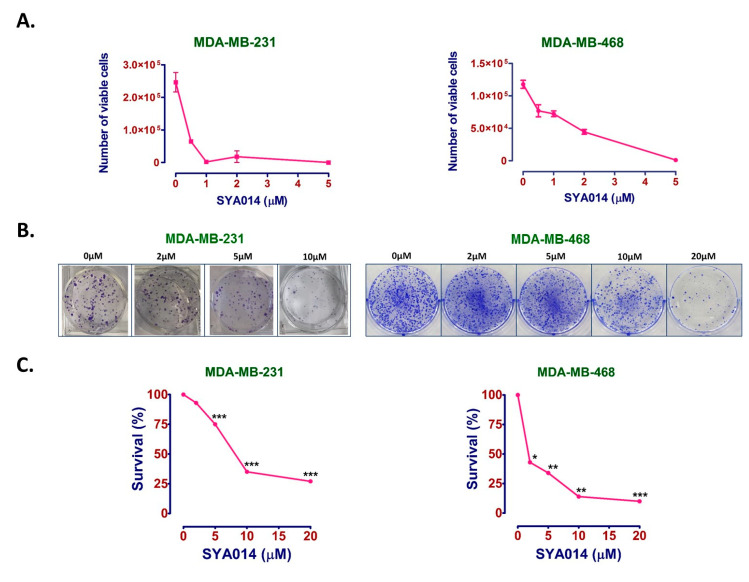
SYA014 inhibits cell proliferation and colony formation of TNBC cell lines. SYA014 inhibits the proliferation of (**A**) MDA-MB-231 cells and MDA-MB-468 cells. The compound also inhibits colony formation of both MDA-MB-231 and MDA-MB-468 cells lines. Representative images shown in (**B**). The survival factions resulting from the clonogenic assay were quantified as shown in (**C**) for MDA-MB-231 cells and MDA-MB-468 cells, respectively. * *p* < 0.05, ** *p* < 0.01, and *** *p* < 0.001 vs. control cells were compared by one-way ANOVA analysis followed by the Dunnett’s multiple comparison test.

**Figure 3 cancers-14-06047-f003:**
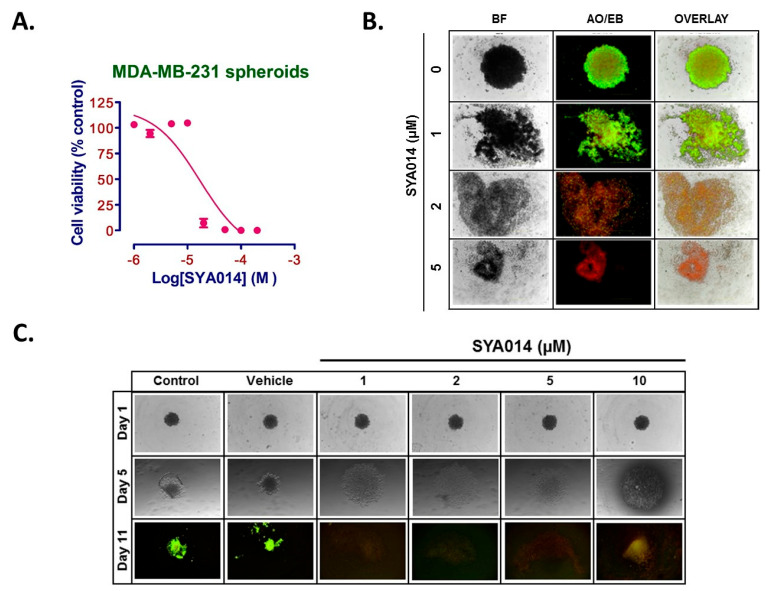
Effect of SYA014 on spheroids of MDA-MB-231 cells. Cell viabilities of MDA-MB-231 spheroids treated with SYA014 (**A**). MDA-MB-231 cells were seeded in 96-Well, U-shaped-bottom Nunclon Sphera plates and incubated for 24 h to form spheroids followed by treatment with SYA014 (0–200 µM) for 24 h and a second treatment for another 24 h (48 h treatment period). The cell viabilities were determined using resazurin reduction assay as described in the materials and methods Section 2.3. Each data point represents the mean ± SEM of 8 wells. SYA014 induces the disintegration of pre-formed spheroids (**B**) and prevents MDA-MB-231 spheroid formation (**C**) of MDA-MB-231 cells upon treatment with SYA014 for 48 h. For the disintegration of pre-formed spheroids, following the SYA014 treatment, spheroids of MDA-MB-231 cells were stained with acridine orange/ethidium bromide (AO/EB). For the inhibition of spheroid formation, on the 11th day, cells were stained with acridine orange/ethidium bromide (AO/EB). Live cells show green color and dead cells in red color. Note: Control and vehicle for day 1, day 5 and day 11 were kept the same while conducting the experiments for SYA014 and our previously published work involving SYA013 [15]. The images for control and vehicle groups are published and are re-used here. BF: Brightfield.

**Figure 4 cancers-14-06047-f004:**
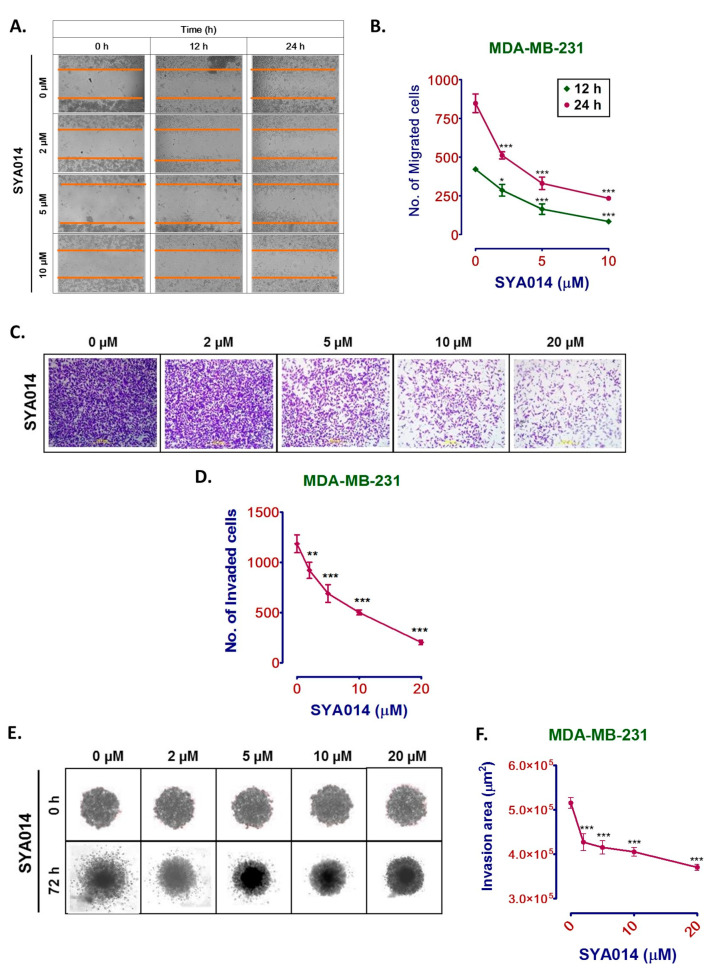
SYA014 inhibits the migration and invasion of MDA-MB-231 cells. MDA-MB-231 cells treated with SYA014 for 48 h inhibit cell migration (**A**) and reduce the number of migrated cells (**B**). The inhibitory effect of SYA014 on MDA-MB-231 cell invasion in the 2D single cell assay (**C**,**D**) and the 3D spheroid invasion assay (**E**,**F**) were determined. Representative images of the crystal violet stained cells after treatment with SYA014 in the 2D invasion assay, and quantification of the invaded cells, are shown in (**C**,**D**), respectively. Representative bright field microscopic images of 3D Spheroid invasion assay and the quantification of the invaded area through the Matrigel compared to the control are shown in **E** and **F**, respectively. Each point represents the mean ± SEM of four determinations. * *p* < 0.05, ** *p* < 0.01, and *** *p* < 0.001 vs. control cells were compared by one-way ANOVA analysis followed by the Dunnett’s multiple comparison test.

**Figure 5 cancers-14-06047-f005:**
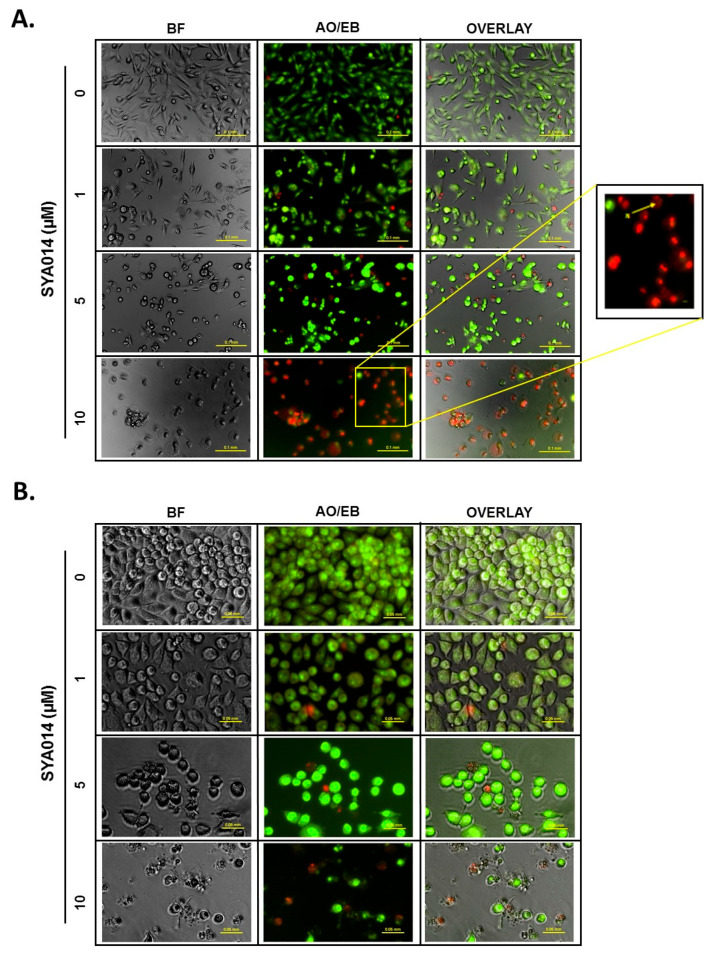
SYA014 induces apoptosis in TNBC cell lines. MDA-MB-231 cells (**A**), and MDA-MB-468 cells (**B**), were treated with SYA014 at different concentrations for 48 h followed by acridine orange/ethidium bromide (AO/EB) dye staining and fluorescence microscopy analysis. Yellow arrow in the magnified image indicate formation of apoptotic bodies. Scale bar = 50 μm. BF: Brightfield.

**Figure 6 cancers-14-06047-f006:**
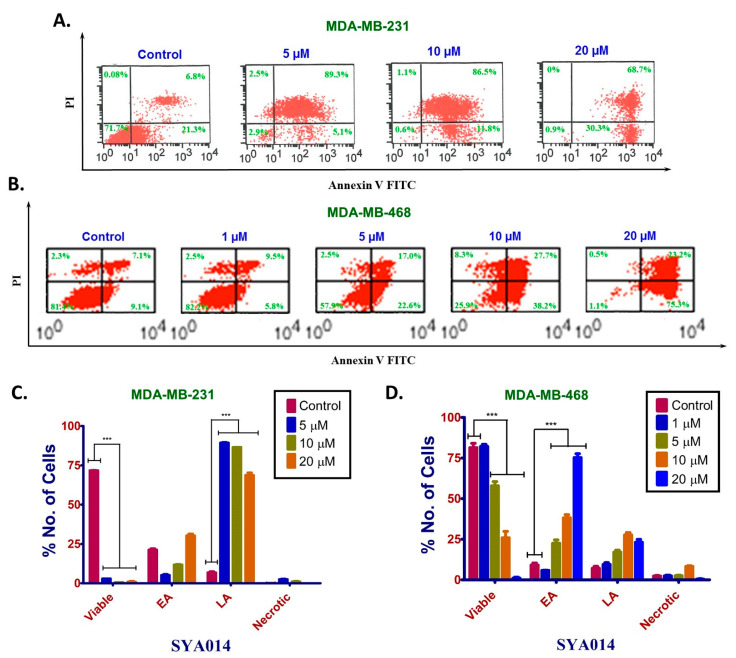
SYA014 triggers cell death by inducing apoptosis in TNBC cell lines. Annexin V and PI staining and flow cytometry analysis was conducted on MDA-MB-231 cells (**A**) and MDA-MB-468 cells (**B**) after treatment with SYA014 (0−20 μM) for 48 h. Data from the flow cytometry analysis for MDA-MB-231 cells and MDA-MB-468 cells were quantified as shown in (**C**,**D**), respectively *** *p* < 0.001. EA = early apoptosis; LA = late apoptosis. No. = number.

**Figure 7 cancers-14-06047-f007:**
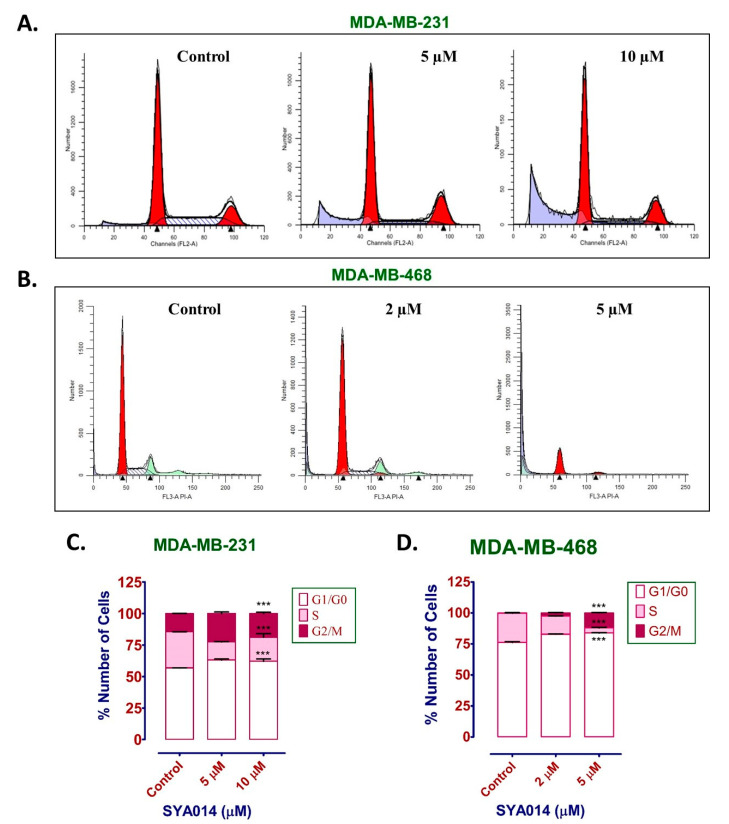
SYA014 arrests cell cycle in TNBC cells. MDA-MB-231 cells (**A**) and MDA-MB-468 cells (**B**) treated with SYA014 at (0–10 µM) and (0–5 µM), respectively, for 48 h, followed by propidium iodide staining and flow cytometry analysis. Quantification of the number of cells at different cell cycle phases for MDA-MB-231 (**C**) and MDA-MB-468 cells (**D**) treated with/without SYA014. *** *p* < 0.001.

**Figure 8 cancers-14-06047-f008:**
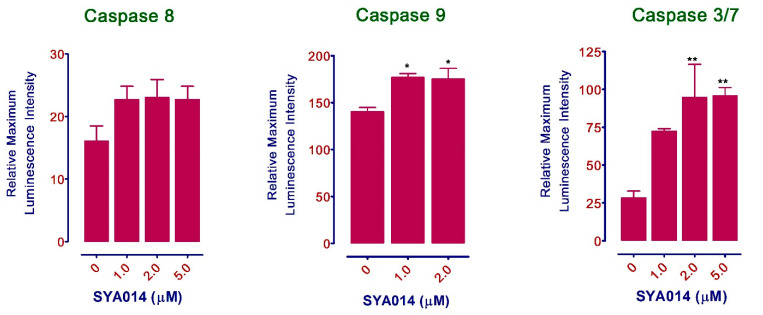
SYA014 induces apoptosis in MDA-MB-231 cells via activation of caspases 3/7 and 9. The activation of caspases 3, 7, and 9 was determined after treating MDA-MB-231 cells with SYA014 for 48 h using Caspase-Glo reagent according to the manufacturer’s protocol. * *p* < 0.05, and ** *p* < 0.01,.

**Figure 9 cancers-14-06047-f009:**
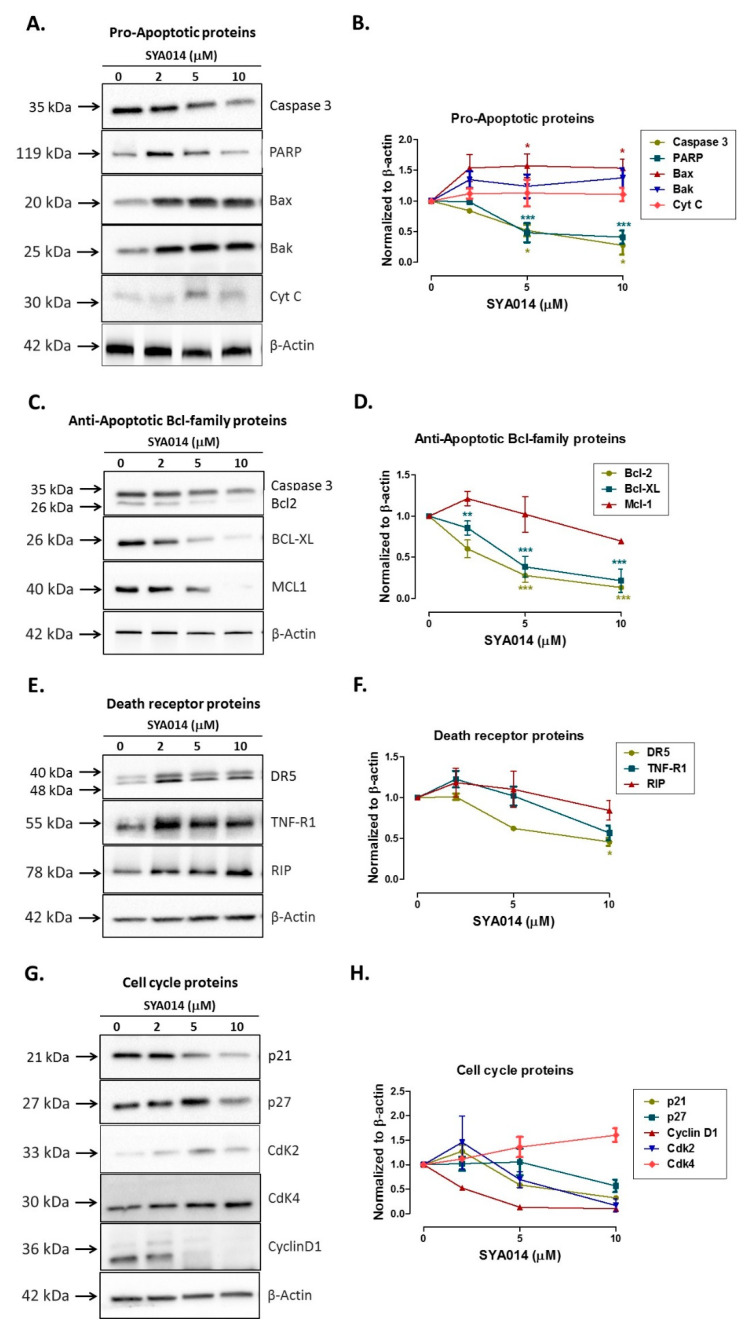
Effect of SYA014 on pro-apoptotic (**A**,**B**), anti-apoptotic (**C**,**D**), death receptor proteins (**E**,**F**), and cell cycle proteins (**G**,**H**). Western blot analyses shows that SYA014 activates the Bcl-2 family pro-apoptotic proteins and antagonizes Bcl-2 family pro-survival proteins in MDA-MB-231 cells. Cells were treated for 48 h and lysed. Aliquots of lysates containing 50 μg of protein were analyzed by western blot, probing with the respective antibodies followed by chemiluminescent detection. β-actin was used as the internal control. Appendix A: Original western blot images. * *p* < 0.05, ** *p* < 0.01, and *** *p* < 0.001.

**Figure 10 cancers-14-06047-f010:**
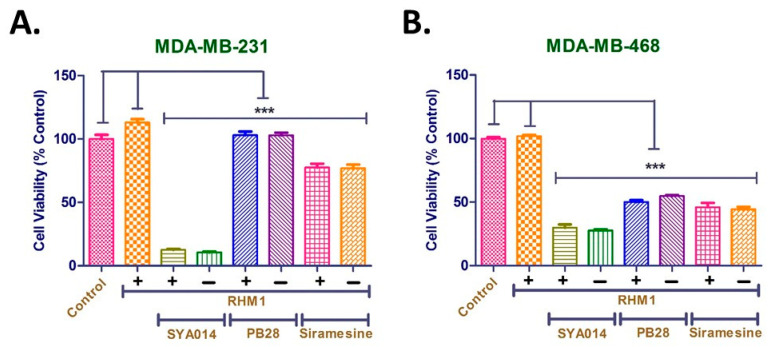
Role of the sigma-2 receptor (σ2R) on the cytotoxic property of SYA014 in MDA-MB-231 cells (**A**) and MDA-MB-468 cells (**B**) pre-treated with RHM-1 followed by the treatment with SYA014 for 24h. The cell viabilities were determined using resazurin reduction assay as described in the materials and methods section. Each experiment was performed in triplicate and the data points represents the mean ± SEM of 8 wells for each experiment. The results were plotted using GraphPad Prism version 5.0. *** *p* <0.001.

**Figure 11 cancers-14-06047-f011:**
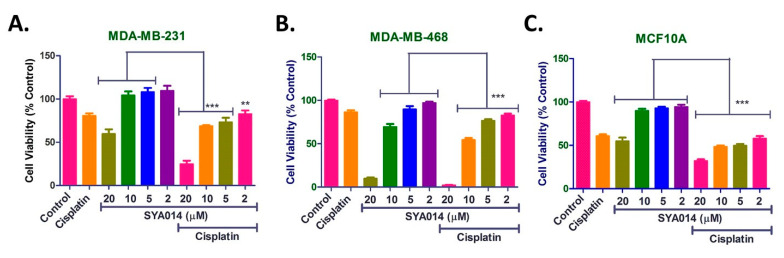
Cytotoxic effects of combination treatment with SYA014 and cisplatin for 72 h in MDA-MB-231 (**A**), MDA-MB-468 (**B**) and MCF10A cells (**C**). The cell viabilities were determined using resazurin reduction assay as described in the materials and methods section. Each experiment was performed in triplicate and the data points represent the mean ± SEM of 6–8 wells for each experiment. The results were plotted using GraphPad Prism version 5.0. ** *p* < 0.01 and *** *p* < 0.001.

## Data Availability

The data presented in this study are available on request from the corresponding author. The data are not publicly available owing to privacy reasons.

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
