# Peer review of "An Evaluation of the Anticancer Properties of SYA014, a Homopiperazine-Oxime Analog of Haloperidol in Triple Negative Breast Cancer Cells"

_cancers, 2022, doi:10.3390/cancers14246047_

Round 1

Reviewer 1 Report (Previous Reviewer 3)

Accept in current form 

Author Response

We thank the reviewers for the valuable comments and suggestions which helped us to improve the manuscript further.

Reviewer 2 Report (Previous Reviewer 1)

The authors have submitted the revised manuscript containing corrections based on the prior review. However, some points raised during the initial review have not yet been addressed within the re-submitted manuscript.

1. Lines 671-673 state "cells treated with SYA014 along with cisplatin did not show any additional toxic effects on MCF10A cells except for SYA014 at 20 μM which showed ~70% cytotoxicity", and lines 757-759 state "SYA014 in combination with cisplatin at a very low concentration (0.5 μM) enhanced the cytotoxic effects in TNBC cell lines while sparing the cell line derived from normal breast epithelial tissue." However, as indicated in Figure 11C, more cytotoxicity occurs in MCF10A cells at all concentrations when cisplatin is combined with SYA014 compared to either cisplatin alone (at 0.5 uM) and compared to SYA014 alone. Thus, the text in both the results and discussion should be revised to match the data shown. The combination did not spare the normal cells from cytotoxicity.

2. Another reviewer from the initial review commented "Discussion needs to be revised, and the presentation of the results and a stronger connection between the contents are needed." However, no additional materials or substantial content have been added to the Discussion section (Section 4) in the re-submitted manuscript. 

3. Section 3.8 mentions figure 12 in a few places, but only figure 11 is present; please ensure all text is corrected to refer to figure 11.

Author Response

This manuscript is a resubmission of an earlier submission. The following is a list of the peer review reports and author responses from that submission.

Round 1

Reviewer 1 Report

This manuscript presents studies in TNBC cells treated with a previously characterized compound that binds to sigma receptors. Prior studies with this compound showed cytotoxic effects in TNBC cells and other malignant cell lines, while the present studies perform more extensive analyses of the extent of cytotoxicity induced by the compound in TNBC cells. This study also determined if blocking sigma 2 receptor with an alternative antagonist changed the cytotoxic effect of the compound in TNBC cells. Several comments are included below for the authors to revise some of the analyses and interpretation of results.

Comments to be addressed:

-Figure 3C presents bright field images from day 1 and 5, but presents fluorescence images from day 11; please include both bright field and fluorescence images at each time point so that proper comparisons can be made.

-Also for Figure 3, the results (section 3.3.2) and caption for Figure 3C state this experiment was done to determine the effect of the compound at blocking spheroid formation, but the data (Figure 3C) show pre-formed spheroids at the beginning of the experiment (which is also stated in the Methods section 2.5). Was the spheroid blocking experimental data shown in Figure 3B, and spheroid disintegration data shown in Figure 3C? Please address this apparent discrepancy between methods and data.

-Figure 9 Western blot data do not seem to match the graphs plotting relative blot intensities for several proteins, particularly the cell cycle proteins (Fig.9G,H); for instance, blots for cyclinD1 and Cdk4 show similar patterns based on concentration of compound used, but the graph presents different trends in normalized ratios for these proteins. Please verify and revise all graphs relative to the data from the respective Western blots, then please update the discussion of the results (section 3.6, and Discussion) to match the updated analyses.

-Figure 10 (graph) is somewhat confusing when indicating which compounds were used in combination. Do the + or – signs indicate RHM1 specifically, while the other compounds were used in all conditions indicated by the lines for those compounds?

-Figure 11 data do not match cytotoxicity data from Figure 1 or other places throughout the manuscript. Figure 11 indicates minimal cytotoxicity from SYA014 below 10 µM in MDA-MB-231 and MDA-MB-468 cells, but all other parts of the manuscript show cytotoxicity from SYA014 as low as 1-2 µM in both cell lines. Is there an explanation for this discrepancy? Also, there appears to be more cytotoxicity in MCF10A cells when SYA014 is combined with cisplatin, so discussion of results (lines 630-635, and Discussion) should be revised to match the data presented in Figure 11C. Finally, numerous studies indicate cisplatin is more cytotoxic against TNBC cells relative to MCF10A cells treated in vitro, but the results from this study indicate cisplatin is more cytotoxic against MCF10A cells than against the TNBC cells treated in this study. This unexpected result should be discussed in light of published literature.

Additional recommended changes:

-line 80: please correct both TNBC cell line designations as “MDA-MB” instead of “MDA-MD”

-please consistently use “cisplatin” instead of the abbreviation “CP”

-lines 485-486: the number of MDA-MB-231 cells in late apoptosis after treatment with SYA014 was not dose-dependent; please revise the wording of this sentence to more closely match the data provided in Figure 6.

-lines 528-530 reference both cell lines, but five of the experiments used one cell line (MDA-MB-231) instead of both cell lines; please revise wording to be more specific about the cells for which data are available

-please be consistent in using the symbol for sigma 2 or writing out “sigma 2” throughout the manuscript

-reference 10 and 14 are the same publication; please ensure all references have correct and completed citation information

Reviewer 2 Report

Gladys M. Asong and colleagues conducted a study entitled "An evaluation of the anticancer properties of SYA014, a Homopiperazine-Oxime analog of haloperidol in Triple Nega-tive Breast Cancer Cells". The topic is interesting, but there are some comments that, if modified, can help improve the quality of the manuscript, which I have listed below:

Be careful to use abbreviations throughout the text (once you introduce, there is no need to re-introduce, write the full name first, then the abbreviation.

The number of these abbreviations in the text is very large, which leads to confusion for the reader.

The text needs to be revised in the native English language person.

Do not use capital letters except for the essentials.

It is suggested not to use long sentences.

What is the reason for choosing two human breast cell lines? Why not use an animal mammary cell line to be useful in vivo studies?

What was the reason for choosing the doses used in SYA014 and cisplatin?

An important and challenging issue is the selection of doses used in each test. IC50 is not visible in this study, and different doses have been used in each study. At the same time, the authors have used the cell viability test.

The quality of all figures is low.

Abbreviations inside the figures should be included in the caption and written in full form.

The scale bar must be specified in microscopic images.

About "3.5.2. SYA014 induces Apoptosis in TNBC cells": What is the reason for choosing different doses in different cell lines? Quadrants are not introduced. What is the importance of the percentage of cells in the Early and Late stages separately? Why are these two stages not combined and not presented under the percentage of apoptosis and necrosis? The diagrams and images obtained from the device need serious revision. What does % No of Cells mean? This title is wrong for charts.

In figure 8, what is the reason for the different doses used?

The discussion needs to be revised, and the presentation of the results and a stronger connection between the contents are needed. Most of the sentences lack references.

You can use the following articles to modify the presentation of the parts mentioned. https://dx.doi.org/10.22128/mch.2022.556.1007 , http://dx.doi.org/10.21203/rs.3.rs-136057/v1

Reviewer 3 Report

The manuscript entitled “An evaluation of the anticancer properties of SYA014, a Homopiperazine-Oxime analog of haloperidol in Triple Negative Breast Cancer Cells” provides new findings. However, the manuscript needs minor editing to make the suitable for publication.

1.      Authors are recommended to include a graphical abstract to summarize the study.

2.       Authors have showed that SYA014 induces apoptosis in TNBC cell lines (MDA-MB-231 and MDA-MB-468 cells). They should include the effect of zVAD after treatment with SYA014 on viability of cells.

3.      In Figure 9A, authors should show cleaved caspase 3 instead if procaspase 3.

4.      Authors should confirm anticancer effects of SYA014 in in vivo model.  
